# Reducing the variance in online optimization by transporting past gradients

**Sébastien M. R. Arnold** *
University of Southern California
Los Angeles, CA
`seb.arnold@usc.edu`

**Pierre-Antoine Manzagol**
Google Brain
Montréal, QC
`manzagop@google.com`

**Reza Babanezhad**
University of British Columbia
Vancouver, BC
`rezababa@cs.ubc.ca`

**Ioannis Mitliagkas**
Mila, Université de Montréal
Montréal, QC
`ioannis@iro.umontreal.ca`

**Nicolas Le Roux**
Mila, Google Brain
Montréal, QC
`nlr@google.com`

## Abstract

Most stochastic optimization methods use gradients once before discarding them. While variance reduction methods have shown that reusing past gradients can be beneficial when there is a finite number of datapoints, they do not easily extend to the online setting. One issue is the staleness due to using past gradients. We propose to correct this staleness using the idea of *implicit gradient transport* (IGT) which transforms gradients computed at previous iterates into gradients evaluated at the current iterate without using the Hessian explicitly. In addition to reducing the variance and bias of our updates over time, IGT can be used as a drop-in replacement for the gradient estimate in a number of well-understood methods such as heavy ball or Adam. We show experimentally that it achieves state-of-the-art results on a wide range of architectures and benchmarks. Additionally, the IGT gradient estimator yields the optimal asymptotic convergence rate for online stochastic optimization in the restricted setting where the Hessians of all component functions are equal.[2]

## 1 Introduction

We wish to solve the following minimization problem:

$$\theta^* = \arg\min_{\theta} E_{x \sim p}[f(\theta, x)] \,, \tag{1}$$

where we only have access to samples $x$ and to a first-order oracle that gives us, for a given $\theta$ and a given $x$, the derivative of $f(\theta, x)$ with respect to $\theta$, i.e. $\frac{\partial f(\theta, x)}{\partial \theta} = g(\theta, x)$. It is known [35] that, when $f$ is smooth and strongly convex, there is a converging algorithm for Problem 1 that takes the form $\theta_{t+1} = \theta_t - \alpha_t g(\theta_t, x_t)$, where $x_t$ is a sample from $p$. This algorithm, dubbed stochastic gradient (SG), has a convergence rate of $O(1/t)$ (see for instance [4]), within a constant factor of the minimax rate for this problem. When one has access to the true gradient $g(\theta) = E_{x \sim p}[g(\theta, x)]$ rather than just a sample, this rate dramatically improves to $O(e^{-\nu t})$ for some $\nu > 0$.

In addition to hurting the convergence speed, noise in the gradient makes optimization algorithms harder to tune. Indeed, while full gradient descent is convergent for constant stepsize $\alpha$, and also

amenable to line searches to find a good value for that stepsize, the stochastic gradient method from [35] with a constant stepsize only converges to a ball around the optimum [38].[3] Thus, to achieve convergence, one needs to use a decreasing stepsize. While this seems like a simple modification, the precise decrease schedule can have a dramatic impact on the convergence speed. While theory prescribes $\alpha_t = O(t^{-\alpha})$ with $\alpha \in (1/2, 1]$ in the smooth case, practitioners often use larger stepsizes like $\alpha_t = O(t^{-1/2})$ or even constant stepsizes.

When the distribution $p$ has finite support, Eq. 1 becomes a finite sum and, in that setting, it is possible to achieve efficient variance reduction and drive the noise to zero, allowing stochastic methods to achieve linear convergence rates [24, 17, 50, 28, 42, 5]. Unfortunately, the finite support assumption is critical to these algorithms which, while valid in many contexts, does not have the broad applicability of the standard SG algorithm. Several works have extended these approaches to the online setting by applying these algorithms while increasing the mini-batch size $N$ [2, 14] but they need to revisit past examples multiple times and are not truly online.

Another line of work reduces variance by averaging iterates [33, 22, 3, 10, 7, 6, 16]. While these methods converge for a constant stepsize in the stochastic case[4], their practical speed is heavily dependent on the fraction of iterates kept in the averaging, a hyperparameter that is thus hard to tune, and they are rarely used in deep learning.

Our work combines two existing ideas and adds a third: a) At every step, it updates the parameters using a weighted average of past gradients, like in SAG [24, 40], albeit with a different weighting scheme; b) It reduces the bias and variance induced by the use of these old gradients by transporting them to "equivalent" gradients computed at the current point, similar to [11]; c) It does so implicitly by computing the gradient at a parameter value different from the current one. The resulting gradient estimator can then be used as a plug-in replacement of the stochastic gradient within any optimization scheme. Experimentally, both SG using our estimator and its momentum variant outperform the most commonly used optimizers in deep learning.

## 2   Momentum and other approaches to dealing with variance

Stochastic variance reduction methods use an average of past gradients to reduce the variance of the gradient estimate. At first glance, it seems like their updates are similar to that of momentum [32], also known as the heavy ball method, which performs the following updates[5]:

$$v_t = \gamma_t v_{t-1} + (1 - \gamma_t) g(\theta_t, x_t), \qquad v_0 = g(\theta_0, x_0)$$
$$\theta_{t+1} = \theta_t - \alpha_t v_t \ .$$

When $\gamma_t = \gamma$, this leads to $\theta_{t+1} = \theta_t - \alpha_t \left( \gamma^t g(\theta_0, x_0) + (1 - \gamma) \sum_{i=1}^{t} \gamma^{t-i} g(\theta_i, x_i) \right)$. Hence, the heavy ball method updates the parameters of the model using an average of past gradients, bearing similarity with SAG [24], albeit with exponential instead of uniform weights.

Interestingly, while momentum is a popular method for training deep networks, its theoretical analysis in the stochastic setting is limited [44], except in the particular setting when the noise converges to 0 at the optimum [26]. Also surprising is that, despite the apparent similarity with stochastic variance reduction methods, current convergence rates are slower when using $\gamma > 0$ in the presence of noise [39], although this might be a limitation of the analysis.

### 2.1   Momentum and variance

We propose here an analysis of how, on quadratics, using past gradients as done in momentum does not lead to a decrease in variance. If gradients are stochastic, then $\Delta_t = \theta_t - \theta^*$ is a random variable. Denoting $\epsilon_i$ the noise at timestep $i$, i.e. $g(\theta_i, x_i) = g(\theta_i) + \epsilon_i$, and writing $\Delta_t - E[\Delta_t] = \alpha \sum_{i=0}^{t} N_{i,t} \epsilon_i$, with $N_{i,t}$ the impact of the noise of the $i$-th datapoint on the $t$-th iterate, we may now analyze the total impact of each $\epsilon_i$ on the iterates. Figure 1 shows the impact of $\epsilon_i$ on $\Delta_t - E[\Delta_t]$ as

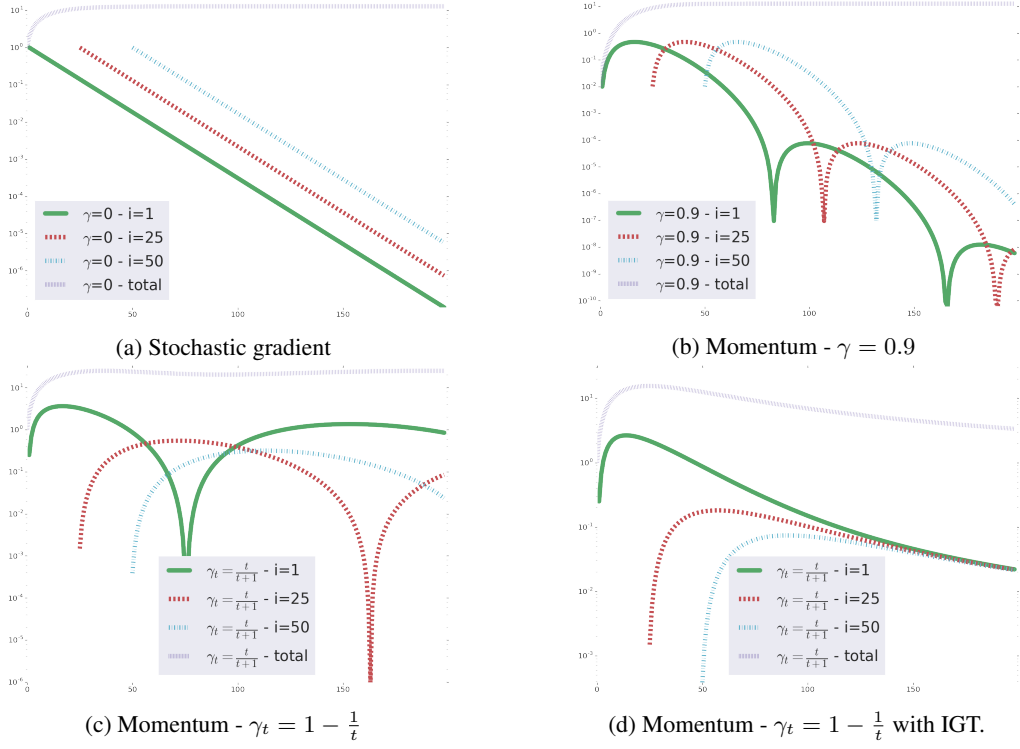

Figure 1: Variance induced over time by the noise from three different datapoints ($i = 1$, $i = 25$ and $i = 50$) as well as the total variance for SG ($\gamma = 0$, *top left*), momentum with fixed $\gamma = 0.9$ (*top right*), momentum with increasing $\gamma_t = 1 - \frac{1}{t}$ without (*bottom left*) and with (*bottom right*) transport. The impact of the noise of each gradient $\epsilon_i$ increases for a few iterations then decreases. Although a larger $\gamma$ reduces the maximum impact of a given datapoint, the total variance does not decrease. With transport, noises are now equal and total variance decreases. The y-axis is on a log scale.

measured by $N_{i,t}^2$ for three datapoints ($i = 1$, $i = 25$ and $i = 50$) as a function of $t$ for stochastic gradient ($\gamma = 0$, left) and momentum ($\gamma = 0.9$, right). As we can see, when using momentum, the variance due to a given datapoint first increases as the noise influences both the next iterate (through the parameter update) and the subsequent updates (through the velocity). Due to the weight $1 - \gamma$ when a point is first sampled, a larger value of $\gamma$ leads to a lower immediate impact of the noise of a given point on the iterates. However, a larger $\gamma$ also means that the noise of a given gradient is kept longer, leading to little or no decrease of the total variance (dashed blue curve). Even in the case of stochastic gradient, the noise at a given timestep carries over to subsequent timesteps, even if the old gradients are not used for the update, as the iterate itself depends on the noise.

At every timestep, the contribution to the noise of the 1st, the 25th and the 50th points in Fig. 1 is unequal. If we assume that the $\epsilon_i$ are i.i.d., then the total variance would be minimal if the contribution from each point was equal. Further, one can notice that the impact of datapoint $i$ is only a function of $t - i$ and not of $t$. This guarantees that the total noise will not decrease over time.

To address these two points, one can increase the momentum parameter over time. In doing so, the noise of new datapoints will have a decreasing impact on the total variance as their gradient is multiplied by $1 - \gamma_t$. Figure 1c shows the impact $N_{i,t}^2$ of each noise $\epsilon_i$ for an increasing momentum $\gamma_t = 1 - \frac{1}{t}$. The peak of noise for $i = 25$ is indeed lower than that of $i = 1$. However, the variance still does not go to 0. This is because, as the momentum parameter increases, the update is an average of many gradients, including stale ones. Since these gradients were computed at iterates already influenced by the noise over previous datapoints, that past noise is amplified, as testified by the higher peak at $i = 1$ for the increasing momentum. Ultimately, increasing momentum does not lead to a convergent algorithm in the presence of noise when using a constant stepsize.

## 2.2 SAG and Hessian modelling

The impact of the staleness of the gradients on the convergence is not limited to momentum. In SAG, for instance, the excess error after $k$ updates is proportional to $\left(1 - \min\left\{\frac{1}{16\widehat{\kappa}}, \frac{1}{8N}\right\}\right)^k$, compared to the excess error of the full gradient method which is $\left(1 - \frac{1}{\kappa}\right)^k$ where $\kappa$ is the condition number of the problem. [6] The difference between the two rates is larger when the minimum in the SAG rate is the second term. This happens either when $\widehat{\kappa}$ is small, i.e. the problem is well conditioned and a lot of progress is made at each step, or when $N$ is large, i.e. there are many points to the training set. Both cases imply that a large distance has been travelled between two draws of the same datapoint.

Recent works showed that correcting for that staleness by modelling the Hessian [46, 11] leads to improved convergence. As momentum uses stale gradients, the velocity is an average of current and past gradients and thus can be seen as an estimate of the true gradient at a point which is not the current one but rather a convex combination of past iterates. As past iterates depend on the noise of previous gradients, this bias in the gradients amplifies the noise and leads to a non-converging algorithm. We shall thus "transport" the old stochastic gradients $g(\theta_i, x_i)$ to make them closer to their corresponding value at the current iterate, $g(\theta_t, x_i)$. Past works did so using the Hessian or an explicit approximation thereof, which can be expensive and difficult to compute and maintain. We will resort to using *implicit transport*, a new method that aims at compensating the staleness of past gradients without making explicit use of the Hessian.

# 3 Converging optimization through implicit gradient transport

Before showing how to combine the advantages of both increasing momentum and gradient transport, we demonstrate how to transport gradients implicitly. This transport is only exact under a strong assumption that will not hold in practice. However, this result will serve to convey the intuition behind implicit gradient transport. We will show in Section 4 how to mitigate the effect of the unsatisfied assumption.

## 3.1 Implicit gradient transport

Let us assume that we received samples $x_0, \ldots, x_t$ in an online fashion. We wish to approach the full gradient $g_t(\theta_t) = \frac{1}{t+1}\sum_{i=0}^{t} g(\theta_t, x_i)$ as accurately as possible. We also assume here that a) We have a noisy estimate $\widehat{g}_{t-1}(\theta_{t-1})$ of $g_{t-1}(\theta_{t-1})$; b) We can compute the gradient $g(\theta, x_t)$ at any location $\theta$. We shall seek a $\theta$ such that

$$\frac{t}{t+1}\widehat{g}_{t-1}(\theta_{t-1}) + \frac{1}{t+1}g(\theta, x_t) \approx g_t(\theta_t) \ .$$

To this end, we shall make the following assumption:

**Assumption 3.1.** *All individual functions $f(\cdot, x)$ are quadratics with the same Hessian $H$.*

This is the same assumption as [10, Section 4.1]. Although it is unlikely to hold in practice, we shall see that our method still performs well when that assumption is violated.

Under Assumption 3.1, we then have (see details in Appendix)

$$g_t(\theta_t) = \frac{t}{t+1}g_{t-1}(\theta_t) + \frac{1}{t+1}g(\theta_t, x_t)$$
$$\approx \frac{t}{t+1}\widehat{g}_{t-1}(\theta_{t-1}) + \frac{1}{t+1}g(\theta_t + t(\theta_t - \theta_{t-1}), x_t) \ .$$

Thus, we can transport our current estimate of the gradient by computing the gradient on the new point at a shifted location $\theta = \theta_t + t(\theta_t - \theta_{t-1})$. This extrapolation step is reminiscent of Nesterov's acceleration with the difference that the factor in front of $\theta_t - \theta_{t-1}$, $t$, is not bounded.

## 3.2 Combining increasing momentum and implicit gradient transport

We now describe our main algorithm, Implicit Gradient Transport (IGT). IGT uses an increasing momentum $\gamma_t = \frac{t}{t+1}$. At each step, when updating the velocity, it computes the gradient of the new point at an extrapolated location so that the velocity $v_t$ is a good estimate of the true gradient $g(\theta_t)$.

We can rewrite the updates to eliminate the velocity $v_t$, leading to the update:

$$\theta_{t+1} = \frac{2t+1}{t+1}\theta_t - \frac{t}{t+1}\theta_{t-1} - \frac{\alpha}{t+1}g\left(\theta_t + t(\theta_t - \theta_{t-1}), x_t\right) \ . \tag{IGT}$$

We see in Fig. 1d that IGT allows a reduction in the total variance, thus leading to convergence with a constant stepsize. This is captured by the following proposition:

**Proposition 3.1.** *If $f$ is a quadratic function with positive definite Hessian $H$ with largest eigenvalue $L$ and condition number $\kappa$ and if the stochastic gradients satisfy: $g(\theta, x) = g(\theta) + \epsilon$ with $\epsilon$ a random i.i.d. noise with covariance bounded by $BI$, then Eq. IGT with stepsize $\alpha = 1/L$ leads to iterates $\theta_t$ satisfying*

$$E[\|\theta_t - \theta^*\|^2] \leq \left(1 - \frac{1}{\kappa}\right)^{2t}\|\theta_0 - \theta^*\|^2 + \frac{d\alpha^2 B\bar{\nu}_0^2}{t} \ ,$$

*with $\nu = (2 + 2\log\kappa)\kappa$ for every $t > 2\kappa$.*

The proof of Prop. 3.1 is provided in the appendix.

Despite this theoretical result, two limitations remain: First, Prop. 3.1 shows that IGT does not improve the dependency on the conditioning of the problem; Second, the assumption of equal Hessians is unlikely to be true in practice, leading to an underestimation of the bias. We address the conditioning issue in the next section and the assumption on the Hessians in Section 4.

## 3.3 IGT as a plug-in gradient estimator

We demonstrated that the IGT estimator has lower variance than the stochastic gradient estimator for quadratic objectives. IGT can also be used as a drop-in replacement for the stochastic gradient in an existing, popular first order method: the heavy ball (HB). This is captured by the following two propositions:

**Proposition 3.2** (Non-stochastic). *In the non-stochastic case, where $B = 0$, variance is equal to $0$ and Heavyball-IGT achieves the accelerated linear rate $O\left(\left(\frac{\sqrt{\kappa}-1}{\sqrt{\kappa}+1}\right)^t\right)$ using the known, optimal heavy ball tuning, $\mu = \left(\frac{\sqrt{\kappa}-1}{\sqrt{\kappa}+1}\right)^2, \alpha = (1 + \sqrt{\mu})^2/L$.*

**Proposition 3.3** (Online, stochastic). *When $B > 0$, there exist constant hyperparameters $\alpha > 0$, $\mu > 0$ such that $\|E[\theta_t - \theta^*]\|^2$ converges to zero linearly, and the variance is $\tilde{O}(1/t)$.*

The pseudo-code can be found in Algorithm 1.

---

**Algorithm 1** Heavyball-IGT

---

1: **procedure** HEAVYBALL-IGT(Stepsize $\alpha$, Momentum $\mu$, Initial parameters $\theta_0$)
2: $\quad$ $v_0 \leftarrow g(\theta_0, x_0)$ $\quad$, $\quad$ $w_0 \leftarrow -\alpha v_0$ $\quad$, $\quad$ $\theta_1 \leftarrow \theta_0 + w_0$
3: $\quad$ **for** $t = 1, \ldots, T-1$ **do**
4: $\quad\quad$ $\gamma_t \leftarrow \frac{t}{t+1}$
5: $\quad\quad$ $v_t \leftarrow \gamma_t v_{t-1} + (1-\gamma_t)g\left(\theta_t + \frac{\gamma_t}{1-\gamma_t}(\theta_t - \theta_{t-1}), x_t\right)$
6: $\quad\quad$ $w_t \leftarrow \mu w_{t-1} - \alpha v_t$
7: $\quad\quad$ $\theta_{t+1} \leftarrow \theta_t + w_t$
8: $\quad$ **end for**
9: $\quad$ **return** $\theta_T$
10: **end procedure**

---

## 4  IGT and Anytime Tail Averaging

So far, IGT weighs all gradients equally. This is because, with equal Hessians, one can perfectly transport these gradients irrespective of the distance travelled since they were computed. In practice, the individual Hessians are not equal and might change over time. In that setting, the transport induces an error which grows with the distance travelled. We wish to average a linearly increasing number of gradients, to maintain the $O(1/t)$ rate on the variance, while forgetting about the oldest gradients to decrease the bias. To this end, we shall use *anytime tail averaging* [23], named in reference to the tail averaging technique used in optimization [16].

Tail averaging is an online averaging technique where only the last points, usually a constant fraction $c$ of the total number of points seen, is kept. Maintaining the exact average at every timestep is memory inefficient and anytime tail averaging performs an approximate averaging using $\gamma_t = \frac{c(t-1)}{1+c(t-1)} \left( 1 - \frac{1}{c} \sqrt{\frac{1-c}{t(t-1)}} \right)$. We refer the reader to [23] for additional details.

## 5  Impact of IGT on bias and variance in the ideal case

To understand the behaviour of IGT when Assumption 3.1 is verified, we minimize a strongly convex quadratic function with Hessian $Q \in \mathbb{R}^{100 \times 100}$ with condition number 1000, and we have access to the gradient corrupted by noise $\epsilon_t$, where $\epsilon_t \sim N(0, 0.3 \cdot I_{100})$. In that scenario where all Hessians are equal and implicit gradient transport is exact, Fig. 2a confirms the $O(1/t)$ rate of IGT with constant stepsize while SGD and HB only converge to a ball around the optimum.

To further understand the impact of IGT, we study the quality of the gradient estimate. Standard stochastic methods control the variance of the parameter update by scaling it with a decreasing stepsize, which slows the optimization down. With IGT, we hope to have a low variance while maintaining a norm of the update comparable to that obtained with gradient descent. To validate the quality of our estimator, we optimized a quadratic function using IGT, collecting iterates $\theta_t$. For each iterate, we computed the squared error between the true gradient and either the stochastic or the IGT gradient. In this case where both estimators are unbiased, this is the trace of the noise covariance of our estimators. The results in Figure 2b show that, as expected, this noise decreases linearly for IGT and is constant for SGD.

We also analyse the direction and magnitude of the gradient of IGT on the same quadratic setup. Figure 2c displays the cosine similarity between the true gradient and either the stochastic or the IGT gradient, as a function of the distance to the optimum. We see that, for the same distance, the IGT gradient is much more aligned with the true gradient than the stochastic gradient is, confirming that variance reduction happens without the need for scaling the estimate.

## 6  Experiments

While Section 5 confirms the performance of IGT in the ideal case, the assumption of identical Hessians almost never holds in practice. In this section, we present results on more realistic and larger scale machine learning settings. All experiments are extensively described in the Appendix A and additional baselines compared in Appendix B.

### 6.1  Supervised learning

**CIFAR10 image classification**   We first consider the task of training a ResNet-56 model [12] on the CIFAR-10 image classification dataset [19]. We use TF official models code and setup [1], varying only the optimizer: SGD, HB, Adam and our algorithm with anytime tail averaging both on its own (ITA) and combined with Heavy Ball (HB-ITA). We tuned the step size for each algorithm by running experiments using a logarithmic grid. To factor in ease of tuning [48], we used Adam's default parameter values and a value of 0.9 for HB's parameter. We used a linearly decreasing stepsize as it was shown to be simple and perform well [43]. For each optimizer we selected the hyperparameter combination that is fastest to reach a consistently attainable target train loss [43]. Selecting the hyperparameter combination reaching the lowest training loss yields qualitatively identical curves. Figure 3 presents the results, showing that IGT with the exponential anytime tail

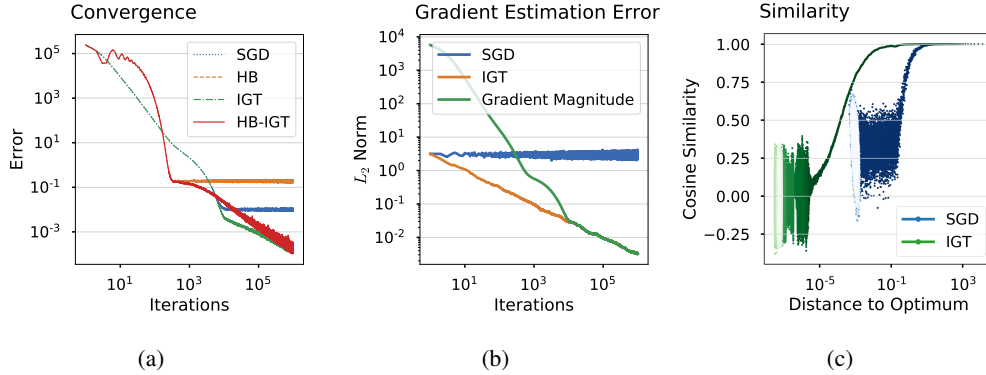

(a)                                      (b)                                      (c)

Figure 2: Analysis of IGT on quadratic loss functions. (a) Comparison of convergence curves for multiple algorithms. As expected, the IGT family of algorithms converges to the solution while stochastic gradient algorithms can not. (b) The blue and orange curves show the norm of the noise component in the SGD and IGT gradient estimates, respectively. The noise component of SGD remains constant, while it decreases at a rate $1/\sqrt{t}$ for IGT. The green curve shows the norm of the IGT gradient estimate. (c) Cosine similarity between the full gradient and the SGD/IGT estimates.

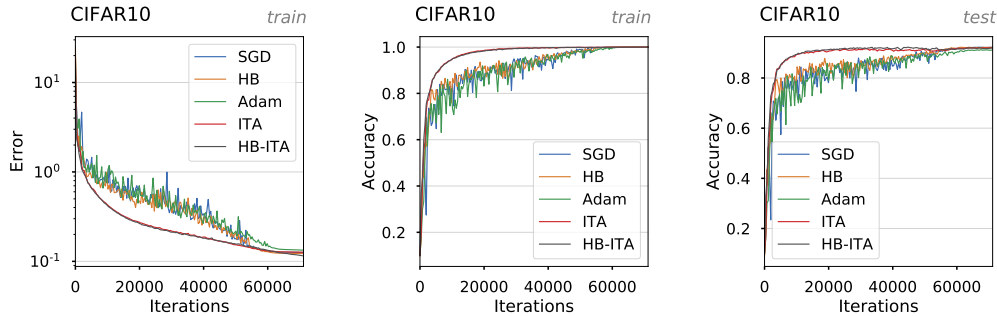

Figure 3: Resnet-56 on CIFAR10. **Left**: Train loss. **Center**: Train accuracy. **Right**: Test accuracy.

average performs favourably, both on its own and combined with Heavy Ball: the learning curves show faster improvement and are much less noisy.

**ImageNet image classification**   We also consider the task of training a ResNet-50 model[12] on the larger ImageNet dataset [36]. The setup is similar to the one used for CIFAR10 with the difference that we trained using larger minibatches (1024 instead of 128). In Figure 4, one can see that IGT is as fast as Adam for the train loss, faster for the train accuracy and reaches the same final performance, which Adam does not. We do not see the noise reduction we observed with CIFAR10, which could be explained by the larger batch size (see Appendix A.1).

**IMDb sentiment analysis**   We train a bi-directional LSTM on the IMDb Large Movie Review Dataset for 200 epochs. [27] We observe that while the training convergence is comparable to HB, HB-ITA performs better in terms of validation and test accuracy. In addition to the baseline and IGT methods, we also train a variant of Adam using the ITA gradients, dubbed **Adam-ITA**, which performs similarly to Adam.

## 6.2   Reinforcement learning

**Linear-quadratic regulator**   We cast the classical linear-quadratic regulator (LQR) [21] as a policy learning problem to be optimized via gradient descent. This setting is extensively described in Appendix A. Note that despite their simple linear dynamics and a quadratic cost functional, LQR systems are notoriously difficult to optimize due to the non-convexity of the loss landscape. [8]

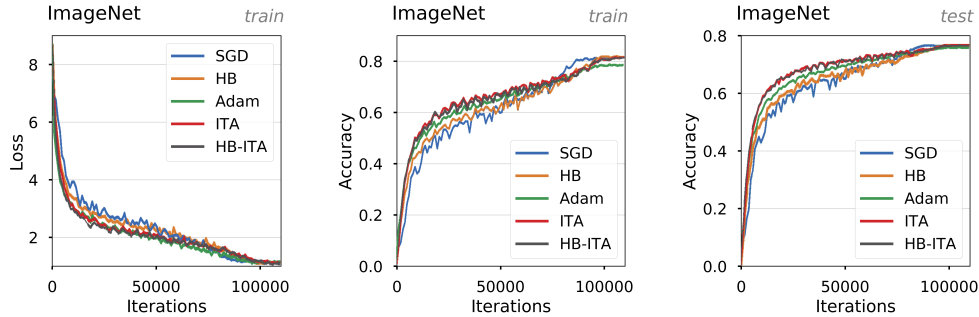

Figure 4: ResNet-50 on ImageNet. **Left**: Train loss. **Center**: Train accuracy. **Right**: Test accuracy.

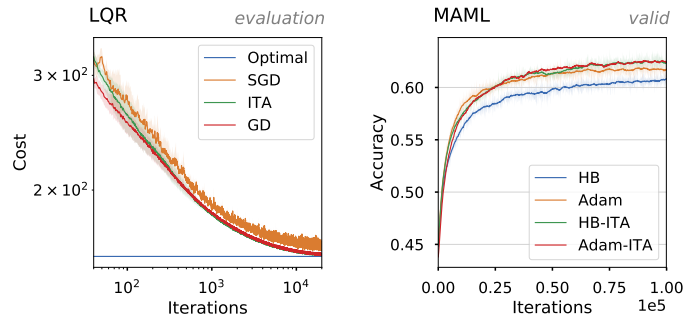

Figure 5: Validation curves for different large-scale machine learning settings. Shading indicates one standard deviation computed over three random seeds. **Left**: Reinforcement learning via policy gradient on a LQR system. **Right**: Meta-learning using MAML on Mini-Imagenet.

The left chart in Figure 5 displays the evaluation cost computed along training and averaged over three random seeds. The first method (**Optimal**) indicates the cost attained when solving the algebraic Riccati equation of the LQR – this is the optimal solution of the problem. **SGD** minimizes the costs using the REINFORCE [47] gradient estimator, averaged over 600 trajectories. **ITA** is similar to SGD but uses the ITA gradient computed from the REINFORCE estimates. Finally, **GD** uses the analytical gradient by taking the expectation over the policy.

We make two observations from the above chart. First, ITA initially suffers from the stochastic gradient estimate but rapidly matches the performance of GD. Notably, both of them converge to a solution significantly better than SGD, demonstrating the effectiveness of the variance reduction mechanism. Second, the convergence curve is smoother for ITA than for SGD, indicating that the ITA iterates are more likely to induce similar policies from one iteration to the next. This property is particularly desirable in reinforcement learning as demonstrated by the popularity of trust-region methods in large-scale applications. [41, 29]

## 6.3 Meta-learning

**Model-agnostic meta-learning** We now investigate the use of IGT in the *model-agnostic meta-learning* (MAML) setting. [9] We replicate the 5 ways classification setup with 5 adaptation steps on tasks from the Mini-Imagenet dataset [34]. This setting is interesting because of the many sources contributing to noise in the gradient estimates: the stochastic meta-gradient depends on the product of 5 stochastic Hessians computed over only 10 data samples, and is averaged over only 4 tasks. We substitute the meta-optimizer with each method, select the stepsize that maximizes the validation accuracy after 10K iterations, and use it to train the model for 100K iterations.

The right graph of Figure 5 compares validation accuracies for three random seeds. We observe that methods from the IGT family significantly outperform their stochastic meta-gradient counter-part, both in terms of convergence rate and final accuracy. Those results are also reflected in the final test

accuracies where Adam-ITA (65.16%) performs best, followed by HB-ITA (64.57%), then Adam (63.70%), and finally HB (63.08%).

## 7 Conclusion and open questions

We proposed a simple optimizer which, by reusing past gradients and transporting them, offers excellent performance on a variety of problems. While it adds an additional parameter, the ratio of examples to be kept in the tail averaging, it remains competitive across a wide range of such values. Further, by providing a higher quality gradient estimate that can be plugged in any existing optimizer, we expect it to be applicable to a wide range of problems. As the IGT is similar to momentum, this further raises the question on the links between variance reduction and curvature adaptation. Whether there is a way to combine the two without using momentum on top of IGT remains to be seen.

**Acknowledgments**

The authors would like to thank Liyu Chen for his help with the LQR experiments and Fabian Pedregosa for insightful discussions.

## Footnotes

*Work done while at Mila.

[2]Open-source implementation available at: `https://github.com/seba-1511/igt.pth`

[3]Under some conditions, it does converge linearly to the optimum [e.g., 45]

[4]Under some conditions on $f$.

[5]This is slightly different from the standard formulation but equivalent for constant $\gamma_t$.

[6]The $\widehat{\kappa}$ in the convergence rate of SAG is generally larger than the $\kappa$ in the full gradient algorithm.

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
