[Supplementary Material · main_with_appendix.pdf]

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

# A  Experimental Details

This section provides additional information regarding the experiments included in the main text.

For each experimental setting we strive to follow the *reproducibility checklist*, and provide:

- a description and citation of the dataset,

- a description of pre-processing steps,

- training / validation / testing splits,

- a description of the hyper-parameter search process and chosen values for each method,

- the exact number of evaluation runs,

- a clear definition of statistics reported, and

- a description of the computing infrastructure.

## A.1  CIFAR10 image classification

**Dataset**   The CIFAR10 dataset [20] consists 50k training and 10k testing images, partitioned over 10 classes. We download and pre-process the images using the TensorFlow `models` package, available at the following URL: `https://github.com/tensorflow/models`

**Model**   We use a residual convolutional network [12] with 56 layers as defined in the `models` package. Specifically, we use the second version whose blocks are built as a batch normalization, then a ReLU activation, and then a convolutional layer. [13]

**Hyper-parameters**   We use the exact setup from `https://github.com/tensorflow/models/officials/resnet`. As such, training is carried out with minibatches of 128 examples for 182 epochs and the training data is augmented with random crops and horizontal flips. Also note this setup multiplies the step size by the size of the minibatch. One deviation from the setup is our use of a linearly decaying learning rate instead of an explicit schedule. The linearly decaying learning rate schedule is simple and was shown to perform well [43]. This schedule is specified using two parameters: the decay rate, a multiplier specifying the final step size (0.1 or 0.01), and the decay step, specifying the step at which the fully decayed rate is reached (always set to 90% of the training steps). To factor in ease of tuning[48] we used Adam's default parameter values and a value of 0.9 for HB's parameter. We used IGT with the exponential Anytime Tail Averaging approach. For the tail fraction, we tried two values: the number of epochs and a tenth of that number (180 and 18). We ran using the following learning rate: (1e0, 3e-1, 1e-1, 3e-2, 1e-2) for SGD, HB and the IGT variants and (1e-2, 3e-3, 1e-3, 3e-4, 1e-4) for Adam. We ran a grid search over the base learning rate and its decay rate with a single run per combination. For each optimizer we selected the hyperparameter combination that is fastest to reach a consistently attainable target train loss of 0.2 [43]. Note that selecting the hyperparameter combination reaching the lowest training loss yields qualitatively identical curves.

The resulting hyper-parameters are:

- SGD stepsize 0.3, decay 0.01

- HB stepsize 0.03, decay 0.01

- Adam stepsize 0.001, decay 0.01

- ITA stepsize 0.3, decay 0.01, tail fraction 18

- HB-ITA stepsize 0.03, decay 0.1, tail fraction 18

**Infrastructure and Runs**   The experiments were run using P100 GPUs (single GPU).

**Additional Results**   We provide all learning curves for the methods comparison presented in the main manuscript in figure 6.

Figure 6: Convergence and accuracy curves along training for the CIFAR10 experiments comparing baseline methods to ours. **Left**: Training. **Right**: Testing.

**Ablation study: importance of IGT correction**   We confirm the importance of the implicit gradient transport correction by running an experiment in which an increasing momentum is used without transport. The results appear in figure 7.

The resulting hyper-parameters are:

- ATA stepsize 0.3, decay 0.01, tail fraction 18

- HB-ATA stepsize 0.03, decay 0.01, tail fraction 18

Figure 7: Convergence and accuracy curves along training for the CIFAR10 experiments comparing the use of ATA combined with our proposed implicit transport mechanism. **Left**: Training. **Right**: Testing.

**Effect of the batch size**    We look into the effect of the batch size. To do so, we plot the number of steps required to reach a reliably attainable training loss of 0.4 as a function of the batch size. We ran using the following mini-batch sizes: 32, 128, 512 and 2048. Running with larger minibatches led to out of memory errors on our single GPU setup. The results presented in figure 8 show the benefit of IGT lowers as the batch size increases. Note that Adam's ability to keep benefiting from larger batch sizes is consistent with previous observations.

## A.2    ImageNet image classification

**Dataset**    We use the 2015 edition of the *ImageNet Large-Scale Visual Recognition Challenge* (ImageNet) [36] dataset. This dataset consists of 1.2M images partitioned into 1'000 classes. We use the pre-processing and loading utilities of the TensorFlow `models` package, available at the following URL: `https://github.com/tensorflow/models`

**Model**    Our model is again a large residual network, consisting of 50 layers. Similar to our CIFAR10 experiments above, we use the implementation described in [13].

**Hyper-parameters**    We used the same setup and approach as for the CIFAR-10 experiments. The setup trains for 90 epochs using mini-batches of 1024 examples. We used a larger grid for the learning rate schedule: decay (0.1, 0.01, 0.001) and decay step fraction (0.7, 0.8, 0.9).

The resulting hyper-parameters are:

CIFAR10

Figure 8: Effect of mini-batch size on the number of steps to reach a target training loss.

- 440    • SGD stepsize 0.3, decay 0.01, decay step 0.8

- 441    • HB stepsize 0.03, decay 0.001, decay step 0.9

- 442    • Adam stepsize 0.0001, decay 0.01, decay step 0.9

- 443    • ITA stepsize 0.3, decay 0.01, tail fraction 90, decay step 0.9

- 444    • HB-ITA stepsize 0.03, decay 0.01, tail fraction 90, decay step 0.9

445  **Infrastructure and Runs**    We ran these experiments using Google TPUv2.

446  **Additional Results**    We include error and accuracy curves for training and testing in Figure 9.

Figure 9: Convergence and accuracy curves along training for the ImageNet experiments. **Left**: Training. **Right**: Testing.

### A.3 IMDb sentiment analysis

**Dataset**  The *Internet Movie Database* (IMDb) [27] consists of 25'000 training and 25'000 test movie reviews. The objective is binary sentiment classification based on the review's text. We randomly split the training set in two folds of 17'536 and 7'552 reviews, the former being used for training and the latter for testing. The data is downloaded, splitted, and pre-processed with `torchtext` package, available at the following URL: `https://github.com/pytorch/text` More specifically, we tokenize the text at the word-level using the `spaCy` package, and embed the tokens using the 100-dimensional GloVe 6B [31] distributed representations.

**Model**  The model consists of an embedding lookup-table, followed by a bi-directional LSTM with dropout, and then by a fully-connected layer. The LSTM uses 256 hidden units and the dropout rate is set to 0.5. The whole model consists of 3.2M trainable parameters, with the embedding lookup-table initilized with the GloVe vectors. The model is trained to minimize the `BCEWithLogitsLoss` with a mini-batch size of 64.

**Hyper-parameters**  For each method, we used a grid-search to find the stepsize minimizing validation error after 15 epochs. The grid starts at 0.00025 and doubles until reaching 0.1024, so as to ensure that no chosen value lies on its boundaries. When applicable, the momentum factor is jointly

optimized over values 0.1 to 0.95. The final hyper-parameters are displayed in the following table for each method.

Table 1: Hyperparameters for IMDb experiments.

|   | HB | Adam | ASGD | HB-IGT | HB-ITA |
|---|---|---|---|---|---|
| $\alpha$ | 0.032 | 0.0005 | 0.064 | 0.128 | 0.064 |
| $\mu$ | 0.95 | 0.95 | n/a | 0.9 | 0.9 |
| $\xi$ | n/a | n/a | 100 | n/a | n/a |
| $\kappa$ | n/a | n/a | $10^5$ | n/a | n/a |

**Infrastructure and Runs**    All IMDb experiments use a single NVIDIA GTX 1080, with PyTorch v0.3.1.post2, CUDA 8.0, and cuDNN v7.0.5. We run each final configurations with 5 different random seeds and always report the mean tendency $\pm$ one standard deviation. Each run lasts approximately three hours and thirty minutes.

**Additional Results**    In addition to the results reported in the main text, we include training, validation, and testing curves for each method in Figure 10. Shading indicates the one standard deviation interval. Note that our focus is explicitly on optimization: in the specific case of IMDb, training for 200 epochs is completely unnecessary from a generalization standpoint as performance degrades rapidly after 15-20 epochs.

Figure 10: Convergence and accuracy curves along training for the IMDb experiments. **Left**: Convergence. **Right**: Accuracy.

## A.4 Linear-quadratic regulator

**Setup**  Our linear-quadratic regulator [21] implements the following equations. The cost functional is evaluated at every timestep $h$ and is given by

$$C(s_h, a_h) = s_h^\top Q s_h + a_h^\top R a_h, \tag{2}$$

for random symmetric positive definite matrices $Q \in \mathbb{R}^{20 \times 20}$ and $R \in \mathbb{R}^{12 \times 12}$ each with condition number 3. The initial state $s_0 \sim \mathcal{N}(0, 3 \cdot I_{20})$ is sampled around the origin, and the subsequent states evolve according to

$$s_{h+1} = As_h + Ba_h, \tag{3}$$

where entries of $A \in \mathbb{R}^{20 \times 20}$, $B \in \mathbb{R}^{20 \times 12}$ are independently sampled from a Normal distribution and then scaled such that the matrix has unit Frobenius norm. The actions are given by the linear stochastic policy $a_h = Ks_h + \epsilon_h^a$, where $\epsilon_h^a \sim \mathcal{N}(0, I)$ and $K$ are the parameters to be optimized.

Gradient methods in this manuscript optimize the sum of costs using the REINFORCE estimate [47] given by

$$\nabla_K \mathbb{E} \sum_h^{10} C(s_h, a_h) = \mathbb{E} \left( \sum_h^{10} \nabla_K \log \pi_K(a_h | s_h) \right) \left( \sum_h^{10} C(s_h, a_h) \right). \tag{4}$$

In our experiments, the above expectation is approximated by the average of 600 trajectory rollouts. Due to the noisy dynamics of the system, it is possible for the gradient norm to explode leading to numerical instabilities – especially when using larger stepsizes. To remedy this issue, we simply discard such problematic trajectories from the gradient estimator.

For each training iteration, we first gather 600 trajectories used for learning and then 600 more used to report evaluation metrics.

**Hyper-parameters**  Due to the simplicity of the considered methods, the only hyper-parameter is the stepsize. For each method, we choose the stepsize from a logarithmically-spaced grid so as to minimize the evaluation cost after 600 iterations on a single seed. Incidentally, the optimal stepsize for GD, SGD, and ITA is 0.0002.

**Infrastructure and Runs**  We use an Intel Core i7-5820K CPU to run the LQR experiments. All methods are implemented using `numpy` v1.15.4. We present results averaged over 3 random seeds, and also report the standard deviation. For stochastic gradient methods (SGD, ITA) training for 20K iterations takes about 3h, for full-gradient method (GD) about 10h, and computing the solution of the Riccati equation takes less than 5 seconds.

**Additional Results**  In addition to the evaluation cost reported in the main text, we also include the cost witnessed during training (and used for optimization) in Figure 11.

Figure 11: LQR costs along training iterations. **Left**: Costs used for learning. **Right**: Costs used for evaluation.

We notice that the training cost curve of ITA is not as smooth as the evaluation one. Similarly, the observed learning costs never reach as good a minima as the evaluation ones. This phenomena is easily clarified: during learning, ITA esimates the gradient using the shifted parameters $K_t + \frac{\gamma_t}{1-\gamma_t}(K_t - K_{t-1})$ as opposed to the true parameters $K_t$. Those shifted parameters are not subject to a reduced variance, hence explaining the observed noise in the cost as well as deteriorated convergence.

## A.5  Model-agnostic meta-learning

**Dataset**    We use the Mini-Imagenet dataset [34] in our model-agnostic meta-learning (MAML) [9] experiments. This dataset comprises 64 training, 12 validation, and 24 test classes. For each of train, validation, and test sets, tasks are constructed by sampling 5 classes from their respective split, and further sampling 5 images per class. Images are downsampled to 84x84x3 tensors of RGB values. For more details, please refer to the official code repository – which we carefully replicated – at the following URL: `https://github.com/cbfinn/maml`

Our implementation departs in two ways from the reference. First, we train our models for 100k iterations as opposed to 60k and only use 5 image samples to compute a meta-gradient whereas the reference implementation uses 15. Second, we only use 5 adaptation steps at evaluation time, when the reference uses 10.

**Model**    The model closely replicates the convolutional neural network of MAML [9]. It consists of 4 layers, each with 32 3x3 kernels, followed by batch normalization and ReLU activations. For specific implementation details, we refer the reader to the above reference implementation.

**Hyper-parameters**    We only tune the meta-stepsize for the MAML experiment. We set the momentum constant to 0.9, the adaptation-stepsize to 0.01, and average the meta-gradient of 4 tasks per iterations. Due to the reduced variance in the gradients, we found it necessary to increase the $\epsilon$ of Adam-ITA to 0.01.

For each method, we search over stepsize values on a logarithmically-spaced grid and select those values that maximize validation accuracy after 10k meta-iterations. These values are reported in Table 2.

| | HB | Adam | HB-ITA | Adam-ITA |
|---|---|---|---|---|
| $\alpha$ | 0.008 | 0.001 | 0.008 | 0.0005 |

Table 2: Stepsizes for MAML experiments.

**Infrastructure and Runs** Each MAML experiment is run on a single NVIDIA GTX TITAN X, with PyTorch v1.1.0, CUDA 8.0, and cuDNN v7.0.5. We run each configuration with 3 different random seeds and report the mean tendency $\pm$ one standard deviation. Each run takes approximately 36 hours, and we evaluate the validation and testing accuracy every 100 iteration.

**Additional Results** We complete the MAML validation curves from the main manuscript with training and testing accuracy curves in Figure 12. Moreover, we recall the final test accuracies for each method: Adam-ITA reaches 65.16%, HB-ITA 64.57%, Adam 63.70%, and HB 63.08%.

Figure 12: Training, validation, and testing accuracies for the MAML experiments along training. Shading indicates the 1 standard deviation interval. **Left**: Training. **Center**: Validation. **Right**: Testing.

# B Additional Experiments

This section presents additional experiments to the ones included in the main text.

## B.1 Baselines comparisons

While experiments in Section 5 highlighted properties of IGT and HB-IGT when the assumption of identical, constant Hessians was verified, we now turn to more realistic scenarios where individual functions are neither quadratic nor have the same Hessian to compare our proposed methods against popular baselines for the online stochastic optimization setting. We target optimization benchmarks and focus on training loss minimization.

Figure 13: Training loss curves for different optimization algorithms on several popular benchmarks. For each method, the hyper-parameters are tuned to minimize the training error after 15 epochs. Algorithms using the IGT gradient estimates tend to outperform their stochastic gradient counter-parts. **Left**: Logistic regression on MNIST. **Center**: LeNet5 on MNIST. **Right**: MobileNetv2 on CIFAR10.

We investigate three different scenarios: (a) **linear-mnist**: a logistic regression model on MNIST, (b) **mnist**: a modified version of LeNet5 [25] on MNIST and (c) **cifar-small**: the MobileNetv2

|               | Linear-MNIST   | MNIST          | CIFAR10        | IMDb           |
| ------------- | -------------- | -------------- | -------------- | -------------- |
| Heavyball     | $92.52 \pm 0.04$ | $99.08 \pm 0.07$ | $91.55 \pm 0.25$ | $86.90 \pm 0.67$ |
| Adam          | $92.57 \pm 0.10$ | $98.99 \pm 0.05$ | $89.36 \pm 0.75$ | $85.62 \pm 0.63$ |
| ASGD          | $92.47 \pm 0.08$ | $99.15 \pm 0.07$ | $91.45 \pm 0.20$ | $87.31 \pm 0.21$ |
| SVRG          | $92.51 \pm 0.04$ | $99.06 \pm 0.08$ | $86.84 \pm 0.17$ | n/a            |
| SGD-dec       | $92.52 \pm 0.06$ | $99.11 \pm 0.06$ | $87.53 \pm 0.23$ | $86.73 \pm 0.34$ |
| Heavyball-IGT | $92.47 \pm 0.10$ | $99.00 \pm 0.05$ | $12.05 \pm 0.21$ | $86.61 \pm 0.23$ |
| Heavyball-ITA | $92.50 \pm 0.10$ | $99.19 \pm 0.02$ | $90.37 \pm 0.31$ | $87.26 \pm 0.24$ |

Table 3: Test accuracies from the best validation epoch.

architecture [37] consisting of 19 convolutional layers on CIFAR10. All models are trained with a mini-batch size of 64, while the remaining hyper-parameters are available in Tables 4, 5, and 6.

For each of the above tasks, models are trained for 200 epochs. We compare the following methods:

- **HB**: the heavy ball method [33],

- **Adam** [18],

- **ASGD** [15],

- **SVRG** [17],

- **SGD-dec**: stochastic gradient method with an exponential learning rate schedule and exponential constant 0.999,

- **HB-IGT**: the heavy ball using the IGT as a plug-in estimator, and

- **HB-ITA**: same as HB-IGT but using the anytime tail averaging to forget the oldest gradients.

The hyperparameters of each method, and in particular the stepsize, are tuned independently according to a logarithmic grid so as to minimize the mean training error after epoch 15 on one seed. We then use those parameters on 5 random seeds and report the mean and standard deviation of the performance.

Figure 13 shows the training curves for the five algorithms in the three settings.

First, we observe that, for the logistic regression, HB-IGT performs on par with HB-ITA and far better than all the other methods, even though the assumption on the Hessians is violated. When using a ConvNet, however, we see that HB-IGT is not competitive with state-of-the-art methods such as Adam or ASGD. HB-ITA, on the other hand, with its smaller reliance on the assumption, once again performs much better than all other methods. In fact, HB-ITA not only converges to a lower final train error but also has a faster initial rate.

While our focus is on optimization, we also report generalization metrics in Table 3. For each algorithm, we computed the best mean accuracy after each epoch on the test set and report this value together with its standard deviation. The importance of the Anytime Tail-Averaging mechanism is again apparent: without it, Heavyball-IGT is unable to improve for more than a few epochs on the CIFAR10 validation set, regardless of the stepsize choice. On the other hand, it is evident from those results that the solutions found by Heavyball-ITA are competitive with the ones discovered by other optimization algorithms.

|         | HB     | Adam    | ASGD    | HB-IGT  | HB-ITA  |
| ------- | ------ | ------- | ------- | ------- | ------- |
| $\alpha$ | 0.0128 | 0.0002  | 0.0032  | 0.0032  | 0.0016  |
| $\mu$   | 0.1    | 0.95    | n/a     | 0.9     | 0.1     |
| $\xi$   | n/a    | n/a     | 10      | n/a     | n/a     |
| $\kappa$ | n/a    | n/a     | $10^4$  | n/a     | n/a     |

Table 4: Hyperparameters for linear-MNIST experiments.

| | HB | Adam | ASGD | HB-IGT | HB-ITA |
|---|---|---|---|---|---|
| $\alpha$ | 0.0064 | 0.0016 | 0.0128 | 0.0032 | 0.0032 |
| $\mu$ | 0.9 | 0.95 | n/a | 0.95 | 0.95 |
| $\xi$ | n/a | n/a | 10 | n/a | n/a |
| $\kappa$ | n/a | n/a | $10^4$ | n/a | n/a |

Table 5: Hyperparameters for MNIST experiments.

| | HB | Adam | ASGD | HB-IGT | HB-ITA |
|---|---|---|---|---|---|
| $\alpha$ | 0.0512 | 0.0512 | 0.1024 | 0.0128 | 0.0512 |
| $\mu$ | 0.95 | 0.9 | n/a | 0.9 | 0.1 |
| $\xi$ | n/a | n/a | 100 | n/a | n/a |
| $\kappa$ | n/a | n/a | $10^5$ | n/a | n/a |

Table 6: Hyperparameters for MobileNetV2 on CIFAR10 experiments.

## C  Proofs

### C.1  Transport formula

$$
\begin{aligned}
g_t(\theta_t) &= \frac{t}{t+1} g_{t-1}(\theta_t) + \frac{1}{t+1} g(\theta_t, x_t) \\
&= \frac{t}{t+1} \left( g_{t-1}(\theta_{t-1}) + H(\theta_t - \theta_{t-1}) \right) + \frac{1}{t+1} g(\theta_t, x_t) && \text{(Quadratic } f) \\
&= \frac{t}{t+1} g_{t-1}(\theta_{t-1}) + \frac{1}{t+1} \left( g(\theta_t, x_t) + tH(\theta_t - \theta_{t-1}) \right) \\
&= \frac{t}{t+1} g_{t-1}(\theta_{t-1}) + \frac{1}{t+1} g(\theta_t + t(\theta_t - \theta_{t-1}), x_t) && \text{(Identical Hessians)} \\
&\approx \frac{t}{t+1} \widehat{g}_{t-1}(\theta_{t-1}) + \frac{1}{t+1} g(\theta_t + t(\theta_t - \theta_{t-1}), x_t) . && (\widehat{g}_{t-1} \text{ is an approximation)}
\end{aligned}
$$

## D  Proof of Prop. 3.1

In this proof, we assume that $g$ is a strongly-convex quadratic function with hessian $H$.

At timestep $t$, we have access to a stochastic gradient $g(\theta, x_t) = g(\theta_t) + \epsilon_t$ where the $\epsilon_t$ are i.i.d. with variance $C \preceq \sigma^2 H$.

We first prove a simple lemma:

**Lemma D.1.** *If $v_0 = g(\theta_0) + \epsilon_0$ and, for $t > 0$, we have*

$$
v_t = \frac{t}{t+1} v_{t-1} + \frac{1}{t+1} g\left( \theta_t + t(\theta_t - \theta_{t-1}) \right) + \frac{1}{t+1} \epsilon_t ,
$$

*then*

$$
v_t = g(\theta_t) + \frac{1}{t+1} \sum_{i=0}^{t} \epsilon_i .
$$

583  *Proof.* Per our assumption, this is true for $t = 0$. Now let us prove the result by induction. Assume
584  this is true for $t - 1$. Then we have:

$$
\begin{aligned}
v_t &= \frac{t}{t+1} v_{t-1} + \frac{1}{t+1} g(\theta_t + t(\theta_t - \theta_{t-1})) + \frac{1}{t+1} \epsilon_t \\
&= \frac{t}{t+1} g(\theta_{t-1}) + \frac{1}{t+1} \sum_{i=0}^{t-1} \epsilon_i \\
&\quad + \frac{1}{t+1} g(\theta_t + t(\theta_t - \theta_{t-1})) + \frac{1}{t+1} \epsilon_t && \text{(recurrence assumption)} \\
&= \frac{t}{t+1} g(\theta_{t-1}) + \frac{1}{t+1} \sum_{i=0}^{t-1} \epsilon_i \\
&\quad + g(\theta_t) - \frac{t}{t+1} g(\theta_{t-1}) + \frac{1}{t+1} \epsilon_t && \text{($g$ is quadratic)} \\
&= g(\theta_t) + \frac{1}{t+1} \sum_{i=0}^{t} \epsilon_i \ .
\end{aligned}
$$

585  This concludes the proof. $\qquad\square$

586  **Lemma D.2.** *Let us assume we perform the following iterative updates:*

$$
\begin{aligned}
v_t &= \frac{t}{t+1} v_{t-1} + \frac{1}{t+1} g(\theta_t + t(\theta_t - \theta_{t-1})) + \frac{1}{t+1} \epsilon_t \\
\theta_{t+1} &= \theta_t - \alpha v_t \ ,
\end{aligned}
$$

587  *starting from $v_0 = g(\theta_0) + \epsilon_0$. Then, denoting $\Delta_t = \theta_t - \theta^*$, we have*

$$
\Delta_t = (I - \alpha H)^t \Delta_0 - \alpha \sum_{i=0}^{t-1} N_{i,t} \epsilon_i
$$

588  *with*

$$
\begin{aligned}
N_{i,0} &= 0 \\
N_{i,t} &= (I - \alpha H) N_{i,t-1} + 1_{i<t} \frac{1}{t} I \ .
\end{aligned}
$$

589  *Proof.* The result is true for $t = 0$. We now prove the result for all $t$ by induction. Let us assume this
590  is true for $t - 1$. Using Lemma D.1, we have

$$
v_{t-1} = g(\theta_{t-1}) + \frac{1}{t} \sum_{i=0}^{t-1} \epsilon_i
$$

591  and thus, using $g(\theta_{t-1}) = H \Delta_{t-1}$,

$$
\begin{aligned}
\Delta_t &= \Delta_{t-1} - \alpha v_{t-1} \\
&= \Delta_{t-1} - \alpha H \Delta_{t-1} - \frac{\alpha}{t} \sum_{i=0}^{t-1} \epsilon_i \\
&= (I - \alpha H) \Delta_{t-1} - \frac{\alpha}{t} \sum_{i=0}^{t-1} \epsilon_i \\
&= (I - \alpha H)^t \Delta_0 - \alpha \sum_{i=0}^{t-2} (I - \alpha H) N_{i,t-1} \epsilon_i - \frac{\alpha}{t} \sum_{i=0}^{t-1} \epsilon_i && \text{(recurrence assumption)} \\
&= (I - \alpha H)^t \Delta_0 - \alpha \sum_{i=0}^{t-1} N_{i,t} \epsilon_i
\end{aligned}
$$

with

$$N_{i,t} = (I - \alpha H)N_{i,t-1} + 1_{i<t}\frac{1}{t}I .$$

This concludes the proof. $\qquad\square$

For the following lemma, we will assume that the Hessian is diagonal and will focus on one dimension with eigenvalue $h$. Indeed, we know that there are no interactions between the eigenspaces and that we can analyze each of them independently [30].

**Lemma D.3.** *Denote $r_h = 1 - \alpha h$. We assume $\alpha \leq \frac{1}{L}$. Then, for any $i$ and any $t$, we have*

$$
\begin{aligned}
&N_{i,t} \geq 0 &\text{(Positivity)}\\
&N_{i,t} = 0 \quad \textit{if } t \leq i &\text{(Zero-start)}\\
&N_{i,t} \leq \log\left(\frac{2}{i(1-r_h)}\right) \quad \textit{if } i < t \leq \frac{2}{1-r_h)} &\text{(Constant bound)}\\
&N_{i,t} \leq \frac{\max\left\{1+r_h, 2\log\left(\frac{2}{i(1-r_h)}\right)\right\}}{t(1-r_h)} \quad \textit{if } \frac{2}{1-r_h} \leq t . &\text{(Decreasing bound)}
\end{aligned}
$$

*Proof.* The Zero-start case $i \geq t$ is immediate from the recursion of Lemma D.2. The Positivity property of $N_{i,t}$ is also immediate from the recursion since the stepsize $\alpha$ is such that $r_h = 1 - \alpha h$ is positive.

We now turn to the Constant bound property. We have, for $t > i$,

$$
\begin{aligned}
N_{i,t} &= r_h N_{i,t-1} + \frac{1}{t}\\
&\leq N_{i,t-1} + \frac{1}{t} .
\end{aligned}
$$

Thus, $N_{i,t} - N_{i,t-1} \leq \frac{1}{t}$. Summing these inequalities, we get a telescopic sum and, finally:

$$
\begin{aligned}
N_{i,t} &\leq \sum_{j=i+1}^{t} \frac{1}{j}\\
&\leq \int_{x=i}^{t} \frac{dx}{x}\\
&= \log\left(\frac{t}{i}\right) .
\end{aligned}
$$

This bound is trivial in the case $i = 0$. In that case, we keep the first term in the sum separate and get

$$N_{0,t} \leq 1 + \log t .$$

In the remainder, we shall keep the $\log\left(\frac{t}{i}\right)$ bound for simplicity. The upper bound on the right-hand size is increasing with $t$ and its value for $t = \frac{2}{1-r_h}$ is thus an upper bound for all smaller values of $t$. Replacing $t$ with $\frac{2}{1-r_h}$ leads to

$$
\begin{aligned}
N_{i,\frac{2}{1-r_h}} &\leq \log\left(\frac{\frac{2}{1-r_h}}{i}\right)\\
&= \log\left(\frac{2}{i(1-r_h)}\right) .
\end{aligned}
$$

This proves the third inequality.

We shall now prove the Decreasing bound by induction. This bound states that, for $t$ large enough, each $N_{i,t}$ decreases as $O(1/t)$. Using the second and third inequalities, we have

$$N_{i,\frac{2}{1-r_h}} \leq \log\left(\frac{2}{i(1-r_h)}\right)^{\frac{2}{1-r_h}} \frac{2}{\frac{2}{1-r_h}}$$

$$= \frac{\log\left(\frac{2}{i(1-r_h)}\right)}{1-r_h} \frac{2}{\frac{2}{1-r_h}}$$

$$\leq \frac{\max\left\{1+r_h, 2\log\left(\frac{2}{i(1-r_h)}\right)\right\}}{\frac{2}{1-r_h}(1-r_h)}.$$

The maximum will help us prove the last property. Thus, for $t = \frac{2}{1-r_h}$, we have

$$N_{i,t} \leq \frac{\max\left\{1+r_h, 2\log\left(\frac{1}{i(1-r_h)}\right)\right\}}{t(1-r_h)}$$

$$\leq \frac{\nu_i}{t},$$

with $\nu_i = \frac{\max\left\{1+r_h, 2\log\left(\frac{1}{i(1-r_h)}\right)\right\}}{(1-r_h)}$. The Decreasing bound is verified for $t = \frac{2}{1-r_h}$.

We now show that if, for any $t > \frac{2}{1-r_h}$, we have $N_{i,t-1} \leq \frac{\nu_i}{t-1}$, then $N_{i,t} \leq \frac{\nu_i}{t}$. Assume that there is such at $t$. Then

$$N_{i,t} = r_h N_{i,t-1} + \frac{1}{t}$$

$$\leq \frac{r_h \nu_i}{t-1} + \frac{1}{t}$$

$$= \frac{r_h t \nu_i + t - 1}{t(t-1)}$$

$$= \frac{(t-1)\nu_i + (r_h - 1)t\nu_i + \nu_i + t - 1}{t(t-1)}$$

$$= \frac{\nu_i}{t} + \frac{(r_h - 1)t\nu_i + \nu_i + t - 1}{t(t-1)}.$$

We now shall prove that $(r_h - 1)t\nu_i + \nu_i + t - 1 = [(r_h - 1)\nu_i + 1]t + \nu_i - 1$ is negative. First, we have that

$$(r_h - 1)\nu_i + 1 = 1 - \max\left\{1 + r_h, 2\log\left(\frac{1}{i(1-r_h)}\right)\right\}$$

$$\leq 0.$$

Then,

$$[(r_h - 1)\nu_i + 1]t + \nu_i - 1 \leq 0 \iff t \geq \frac{\nu_i - 1}{(1-r_h)\nu_i - 1}$$

since $(r_h - 1)\nu_i + 1 \leq 0$. Thus, the property is true for every $t \geq \frac{\nu_i - 1}{(1-r_h)\nu_i - 1}$. In addition, we have

$$\nu_i \geq \frac{1 + r_h}{1 - r_h}$$

$$\nu_i(1 - r_h) \geq 1 + r_h$$

$$2\nu_i(1 - r_h) - 2 \geq \nu_i(1 - r_h) - 1 + r_h$$

$$\frac{2}{1 - r_h} \geq \frac{\nu_i - 1}{\nu_i(1 - r_h) - 1},$$

and the property is also true for every $t \geq \frac{2}{1-r_h}$. This concludes the proof. $\square$

619    Finally, we can prove the Proposition 3.1:

620    *Proof.* The expectation of $\Delta_t$ is immediate using Lemma D.2 and the fact that the $\epsilon_i$ are independent,
621    zero-mean noises. The variance is equal to $V[\Delta_t] = \alpha^2 B \sum_{i=0}^{t} N_{i,t}^2$. While our analysis was
622    only along one eigenspace of the Hessian with associated eigenvalue $h$, we must now sum over all
623    dimensions. We will thus define

$$\bar{\nu}_i = \frac{\max\left\{2 - \alpha\mu, 2\log\left(\frac{1}{i\alpha\mu}\right)\right\}}{\alpha\mu} \quad \text{for } i > 0$$

$$\bar{\nu}_0 = \frac{2 + 2\log\left(\frac{1}{\alpha\mu}\right)}{\alpha\mu} \ ,$$

624    which is, for every $i$, the maximum $\nu_i$ across all dimensions. We get

$$V[\Delta_t] \leq d\alpha^2 B \sum_{i=0}^{t} \frac{\bar{\nu}_i^2}{t^2}$$

$$\leq d\alpha^2 B \sum_{i=0}^{t} \frac{\bar{\nu}_0^2}{t^2} \quad \text{since } \nu_i \geq \nu_{i+1} \ \forall i$$

$$\leq \frac{d\alpha^2 B \bar{\nu}_0^2}{t} \ .$$

625    Since we have

$$E[\theta_t - \theta^*] = (I - \alpha H)^t (\theta_0 - \theta^*) \ ,$$

626    we get

$$E[\|\theta_t - \theta^*\|^2] = \|E[\theta_t - \theta^*]\|^2 + V[\Delta_t]$$

$$\leq (\theta_0 - \theta^*)^\top (I - \alpha H)^{2t} (\theta_0 - \theta^*) + \frac{d\alpha^2 B \bar{\nu}_0^2}{t}$$

$$\leq \left(1 - \frac{1}{\kappa}\right)^{2t} \|\theta_0 - \theta^*\|^2 + \frac{d\alpha^2 B \bar{\nu}_0^2}{t} \ .$$

627    This concludes the proof. $\qquad\qquad\square$

# E    Proof of Proposition 3.2 and Proposition 3.3

629    In this section we list and prove all lemmas used in the proofs of Proposition 3.2 and Proposition 3.3;
630    all lemmas are stated in the same conditions as the proposition.

631    We start the following proposition:

632    **Proposition E.1.** *Let $f$ be a quadratic function with positive definite Hessian $H$ with largest eigen-*
633    *value $L$ and condition number $\kappa$ and if the stochastic gradients satisfy $g(\theta, x) = g(\theta) + \epsilon$ with $\epsilon$ a*
634    *random uncorrelated noise with covariance bounded by $BI$.*

635    *Then, Algorithm 1 leads to iterates $\theta_t$ satisfying*

$$E[\theta_t - \theta^*] = \begin{pmatrix} I \\ 0 \end{pmatrix} A^t \begin{pmatrix} E[\theta_1 - \theta^*] \\ E[\theta_0 - \theta^*] \end{pmatrix} \tag{5}$$

636    *where*

$$A = \begin{pmatrix} I - \alpha H + \mu I & -\mu I \\ I & 0 \end{pmatrix} \tag{6}$$

637    *governs the dynamics of this bias. In particular, when its spectral radius, $\rho(A)$ is less than 1, the*
638    *iterates converge linearly to $\theta^*$.*

*In a similar fashion, the variance dynamics of Heavyball-IGT are governed by the matrix*

$$D_i = \begin{pmatrix} (1 - \alpha h_i + \mu)^2 + 2\alpha^2 h_i^2 & \mu^2 & -2\mu(1 - \alpha h_i + \mu)^2 \\ 1 & 0 & 0 \\ 1 - \alpha h_i + \mu & 0 & -\mu \end{pmatrix}$$

*If the spectral radius of $D_i$, $\rho(D_i)$, is strictly less than 1 or all $i$, then there exist constants $t_0 > 0$ and $C > 0$ for which*

$$\mathrm{Var}(\theta_t) \leq 2\alpha^2 dBC \frac{\log(t)}{t}, \quad \text{for } t > t_0$$

*where $B$ is a bound on the variance of noise variables $\epsilon_i$.*

**Lemma E.2** (IGT estimator as true gradient plus noise average). *If $v_0 = g(\theta_0) + \epsilon_0$ and for $t > 0$ we have*

$$v_t = \frac{t}{t+1} v_{t-1} + \frac{1}{t+1} g(\theta_t + t(\theta_t - \theta_{t-1})) + \frac{1}{t+1} \epsilon_t,$$

*then*

$$v_t = g(\theta_t) + \frac{1}{t+1} \sum_{i=0}^{t} \epsilon_i.$$

This lemma is already proved in the previous section for the IGT estimator (Lemma D.1) and is just repeated here for completeness. We will use this result in the next few lemmas.

**Lemma E.3** (The IGT gradient estimator is unbiased on quadratics). *For the IGT gradient estimator, $v_t$, corresponding to parameters $\theta_t$ we have*

$$\mathbb{E}[v_t] = g(\mathbb{E}\theta_t),$$

*where the expectation is over all gradient noise vectors $\epsilon_0, \epsilon_1, \ldots, \epsilon_t$.*

*Proof.* The proof proceeds by induction. The base case holds as we have

$$\mathbb{E}[v_0] = \mathbb{E}[g_0 + \epsilon_0] = g(\theta_0).$$

For the inductive case, we can write

$$\begin{aligned} \mathbb{E}[v_t] &= \mathbb{E}\left[ \frac{t}{t+1} v_{t-1} + \frac{1}{t+1} \hat{g}(\theta_t + t(\theta_t - \theta_{t-1})) \right] \\ &= \mathbb{E}\left[ \frac{t}{t+1} v_{t-1} + \frac{1}{t+1} g_t + \frac{t}{t+1} g_t - \frac{t}{t+1} g_{t-1} + \frac{1}{t+1} \epsilon_t \right] \\ &= \frac{t}{t+1} \mathbb{E}[v_{t-1} - g_{t-1}] + \mathbb{E}[g_t] + \frac{t}{t+1} \mathbb{E}[\epsilon_t] \\ &= \mathbb{E}[g_t] = g(\mathbb{E}[\theta_t]). \end{aligned}$$

Where, in the third equality, $\mathbb{E}[v_{t-1} - g_{t-1}] = 0$ by the inductive assumption, and the last equality because the gradient of a quadratic function is linear. □

**Lemma E.4** (Bounding the IGT gradient variance). *Let $v_t$ be the IGT gradient estimator. Then*

$$\mathrm{Var}[v_t] \leq 2h^2 \mathrm{Var}[\theta_t - \theta^\star] + \frac{2B}{t},$$

*where $B$ is the variance of the homoscedastic noise $\epsilon_t$.*

*Proof.*

$$\mathrm{Var}\left[v_t\right] = \mathrm{Var}\left[g_t + \frac{1}{t+1}\sum_{i=0}^{t}\epsilon_i\right]$$

$$= \mathrm{Var}\left[h\theta_t\right] + \mathrm{Var}\left[\frac{1}{t+1}\sum_{i=0}^{t}\epsilon_i\right]$$

$$+ 2\mathrm{Cov}\left[h\theta_t, \frac{1}{t+1}\sum_{i=0}^{t}\epsilon_i\right]$$

$$\leq 2\mathrm{Var}\left[h\theta_t\right] + 2\mathrm{Var}\left[\frac{1}{t+1}\sum_{i=0}^{t}\epsilon_i\right]$$

$$= 2h^2\mathrm{Var}\left[\theta_t - \theta^\star\right] + 2\frac{B}{t}$$

$\square$

Now that we have these basic results on the IGT estimator, we can analyze the evolution of the bias and variance of Heavyball-IGT. We use the quadratic assumption to decouple the vector dynamics of Heavyball-IGT into independent scalar dynamics. If the Hessian, $H$, has eigenvalues $L \geq h_1 \geq h_2 \geq \ldots \geq h_n = L/\kappa$, then we can assume without loss of generality that $H$ is diagonal with $H_{ii} = h_i$.

**Lemma E.5** (Evolution of bias for scalar quadratic). *Assume that the Hessian, second derivative, is h.*

*Starting with $v_0 = g(\theta_0) + \epsilon_0$ and $w_0 = 0$, performing the following iterative updates (Heavyball-IGT, Algorithm 1):*

$$v_t = \frac{t}{t+1}v_{t-1} + \frac{1}{t+1}g(\theta + t(\theta_t - \theta_{t-1})) + \frac{1}{t+1}\epsilon_t,$$
$$w_{t+1} = \mu w_t + \alpha v_t, \qquad \theta_{t+1} = \theta_t - w_{t+1}$$

*results in*

$$\Delta_t = A^t\Delta_0 - \alpha\sum_{i=0}^{t-1}N_{i,t}\begin{bmatrix}\epsilon_i \\ 0\end{bmatrix}$$

*where $N_{j,0} = 0_{2\times 2}$, $\qquad N_{i,t} = AN_{i,t-1} + 1_{i<t}\frac{1}{t}I$,*

$\Delta_t = \begin{bmatrix}\theta_t - \theta^* \\ \theta_{t-1} - \theta^*\end{bmatrix}$ *and* $A = \begin{pmatrix}1 - \alpha h + \mu & -\mu \\ 1 & 0\end{pmatrix}$.

*Proof.* The proof proceeds by induction. First notice that for $t = 0$ the equality naturally holds. We make the inductive assumption that it holds for $t - 1$, and start by using Lemma E.2:

$$\Delta_t = A\Delta_{t-1} - \frac{\alpha}{t}\sum_{i=0}^{t-1}\begin{bmatrix}\epsilon_i \\ 0\end{bmatrix}$$

$$= A\left(A^{t-1}\Delta_0 - \alpha\sum_{i=0}^{t-2}N_{i,t}\begin{bmatrix}\epsilon_i \\ 0\end{bmatrix}\right) - \frac{\alpha}{t}\sum_{i=0}^{t-1}\begin{bmatrix}\epsilon_i \\ 0\end{bmatrix} \qquad \text{(Inductive assumption)}$$

$$= A^t\Delta_0 - \alpha\left(\sum_{i=0}^{t-2}AN_{i,t}\begin{bmatrix}\epsilon_i \\ 0\end{bmatrix} + \frac{1}{t}\sum_{i=0}^{t-1}\begin{bmatrix}\epsilon_i \\ 0\end{bmatrix}\right)$$

$$= A^t\Delta_0 - \alpha\sum_{i=0}^{t-1}N_{i,t}\begin{bmatrix}\epsilon_i \\ 0\end{bmatrix} \qquad \text{(Def. of } N_{i,t})$$

$\square$

**Lemma E.6** (Evolution of variance). *Let $U_t = \text{Var}\,[\theta_t]$ and $V_t = \text{Cov}\,[\theta_t, \theta_{t-1}]$, where $\theta_t$ is the $t$-th iterate of Heavyball-IGT on a 1-dimensional quadratic function with curvature $h$. The following matrix describes the variance dynamics of Heavyball-IGT.*

$$D = \begin{pmatrix} (1 - \alpha h + \mu)^2 + 2\alpha^2 h^2 & \mu^2 & -2\mu(1 - \alpha h + \mu)^2 \\ 1 & 0 & 0 \\ 1 - \alpha h + \mu & 0 & -\mu \end{pmatrix} \tag{7}$$

*If the spectral radius of $D$, $\rho(D)$, is strictly less than 1, then there exist constants $t_0 > 0$ and $C > 0$ for which*

$$\text{Var}(\theta_t) \le 2\alpha^2 BC \frac{\log(t)}{t}$$

*, where $B$ is a bound on the variance of the noise.*

*Proof.* The proof (and lemma) is similar to the proof of Lemma 9 in [49]. We start by expanding $U_{t+1}$ as follows.

$$
\begin{aligned}
U_{t+1} &= \mathbb{E}\left[ (\theta_{t+1} - \bar{\theta}_{t+1})^2 \right] \\
&= \mathbb{E}\left[ (\theta_t - \alpha v_t + \mu(\theta_t - \theta_{t-1}) - \bar{\theta}_t + \alpha g_t - \mu(\bar{\theta}_t - \bar{\theta}_{t-1}))^2 \right] \\
&= \mathbb{E}[(\theta_t - \alpha g_t + \mu(\theta_t - \theta_{t-1}) - \bar{\theta}_t + \alpha g_t \\
&\quad - \mu(\bar{\theta}_t - \bar{\theta}_{t-1}) + \alpha(g_t - v_t))^2] \\
&= \mathbb{E}\left[ ((1 - \alpha h + \mu)(\theta_t - \bar{\theta}_t) - \mu(\theta_{t-1} - \bar{\theta}_{t-1}))^2 \right] \\
&\quad + \alpha^2 \mathbb{E}\left[ (g_t - v_t)^2 \right] \\
&\le \mathbb{E}\left[ ((1 - \alpha h + \mu)(\theta_t - \bar{\theta}_t) - \mu(\theta_{t-1} - \bar{\theta}_{t-1}))^2 \right] \\
&\quad + \alpha^2 \left( 2h^2 \mathbb{E}\left[ (\theta_t - \bar{\theta}_t)^2 \right] + \frac{2B}{t+1} \right) \\
&\le \left[ (1 - \alpha h + \mu)^2 + 2\alpha^2 \mu^2 \right] \mathbb{E}\left[ (\theta_t - \bar{\theta}_t)^2 \right] \\
&\quad - 2\mu(1 - \alpha h + \mu)\mathbb{E}\left[ (\theta_t - \bar{\theta}_t)(\theta_{t-1} - \bar{\theta}_{t-1}) \right] \\
&\quad + \mu^2 \mathbb{E}\left[ (\theta_{t-1} - \bar{\theta}_{t-1})^2 \right] + \alpha^2 \frac{2B}{t+1}\,.
\end{aligned}
$$

Where the fourth equality is obtained since we know that the IGT gradient estimator is unbiased, i.e. $\mathbb{E}\left[g_t - v_t\right] = 0$. The first inequality stems from Lemma E.4. We similarly expand $V_t$.

$$
\begin{aligned}
V_t &= \mathbb{E}\left[ (\theta_t - \bar{\theta}_t)(\theta_{t-1} - \bar{\theta}_{t-1}) \right] \\
&= \mathbb{E}\left[ ((1 - \alpha h + \mu)(\theta_{t-1} - \bar{\theta}_{t-1}) - \mu(\theta_{t-2} - \bar{\theta}_{t-2}) + \alpha(g_t - v_t)) \right. \\
&\quad \left. (\theta_{t-1} - \bar{\theta}_{t-1}) \right] \\
&= (1 - \alpha h + \mu)\mathbb{E}\left[ (\theta_{t-1} - \bar{\theta}_{t-1})^2 \right] \\
&\quad - \mu \mathbb{E}\left[ (\theta_{t-1} - \bar{\theta}_{t-1})(\theta_{t-2} - \bar{\theta}_{t-2}) \right]
\end{aligned}
$$

From the above expressions, we obtain

$$\begin{pmatrix} U_{t+1} \\ U_t \\ V_{t+1} \end{pmatrix} \le D \begin{pmatrix} U_t \\ U_{t-1} \\ V_t \end{pmatrix} + \begin{pmatrix} \alpha^2 \frac{2B}{t+1} \\ 0 \\ 0 \end{pmatrix}$$

$$\le 2\alpha^2 B \sum_{i=0}^{t} D^i \begin{pmatrix} \frac{1}{t+1-i} \\ 0 \\ 0 \end{pmatrix}$$

$$\le 2\alpha^2 B \left( \sum_{i=0}^{s-1} D^i \begin{pmatrix} \frac{1}{t+1-i} \\ 0 \\ 0 \end{pmatrix} + \sum_{i=s}^{t} D^i \begin{pmatrix} \frac{1}{t+1-i} \\ 0 \\ 0 \end{pmatrix} \right)$$

675    where an inequality of vectors implies the corresponding elementwise inequalities.

676    If the spectral radius of $D$, $\rho(D)$ is strictly less than 1, then there exists constant $C' > 0$ such that

$$\begin{pmatrix} 1 \\ 0 \\ 0 \end{pmatrix}^T \sum_{i=0}^{s-1} D^i \begin{pmatrix} \frac{1}{t+1-i} \\ 0 \\ 0 \end{pmatrix} \le C' \sum_{i=0}^{s-1} \frac{1}{t+1-i}$$

$$\le \frac{C's}{t+2-s}$$

677    If the spectral radius of $D$, $\rho(D)$, is strictly less than 1, then there exists constant $\zeta > 0$ and constant
678    $C''(\zeta) > 0$ such that, $\rho(D) + \zeta < 1$ and

$$\begin{pmatrix} 1 \\ 0 \\ 0 \end{pmatrix}^T \sum_{i=s}^{t} D^i \begin{pmatrix} \frac{1}{t+1-i} \\ 0 \\ 0 \end{pmatrix} \le \begin{pmatrix} 1 \\ 0 \\ 0 \end{pmatrix}^T \sum_{i=s}^{t} D^i \begin{pmatrix} 1 \\ 0 \\ 0 \end{pmatrix}$$

$$\le C'' \sum_{i=s}^{t} (\rho(D) + \zeta)^s$$

$$= C''(t - s + 1)(\rho(D) + \zeta)^s$$

679    Let $\rho' = \rho(D) + \zeta$ and $s = \lceil 2\log_{1/\rho'} t \rceil$. Then $(\rho(D) + \zeta)^s = 1/t^2$, and putting the above two
680    bounds together,

$$U_{t+1} \le 2\alpha^2 B \left( \frac{2C' \log_{1/\rho'} t}{t + 2 - 2\log_{1/\rho'} t} + C'' \frac{t - 2\log_{1/\rho'} t + 1}{t^2} \right)$$

$$\le 2\alpha^2 BC \frac{\log(t+1)}{t+1}$$

681    where the last inequality holds for $t > t_0$ for some $t_0$ and some constant $C > 0$.

682    $\qquad\qquad\qquad\qquad\qquad\qquad\qquad\qquad\qquad\qquad\qquad\qquad\qquad\qquad\qquad$ $\square$

683    We can now prove Proposition E.1.

684    *Proof of Proposition E.1.* The bias statement of the proposition follows directly from taking an
685    expectation on the bound of Lemma B.4, and the variance statement from summing up the $d$ different
686    variance terms given for each scalar component by Lemma B.5. $\qquad\qquad\qquad\qquad$ $\square$

### E.1    Proof of Proposition 3.2

688    This Proposition follows from the observation that, in the noiseless case, $\epsilon_t = 0$ in our model. In that
689    case, Lemma E.3 shows that Heavyball-IGT reduces to the heavy ball, and the rest follows from the
690    optimal tuning of the heavy ball [49].

## E.2 Proof of Proposition 3.3

*Proof.* Like we did in previous proofs, we can assume without loss of generality that the Hessian, $H$, is diagonal with elements $h_i$. For a diagonal $H$, matrix $A$ can be permuted to be block diagonal with blocks

$$A_i = \begin{pmatrix} 1 - \alpha h_i + \mu & \mu \\ 1 & 0 \end{pmatrix}.$$

To prove that $\rho(A) < 1$ it suffices to prove that $\rho(A_i) < 1$ for all $i$. For the rest of the proof we will focus on the dynamics along a single eigendirection with curvature $h_i$. The rest of this proof used $D$ to denote $D_i$, $A$ to denote $A_i$ and $h$ to denote $h_i$.

To make explicit the dependence of matrices $A$ and $D$ on hyperparameters and curvature, we write $A(\alpha, \mu, h)$ and $D(\alpha, \mu, h)$. Let $0 < \alpha < 2/(3h)$ and $\mu_0 = 0$. Using hyperparameters $(\alpha, \mu_0)$ one gets the results for gradient descent without momentum. In particular $\rho(A(\alpha, \mu_0, h)) = |1 - \alpha h| < 1$, and the spectral radius of $D$ is $\rho(D(\alpha, \mu_0, h)) = |(1 - \alpha h)^2 + 2\alpha^2 h^2| < 1$.

We will argue that there exists $\mu > 0$, such that $\rho(A(\alpha, \mu, h)) < 1$, and the spectral radius of $D$ is $\rho(D(\alpha, \mu, h)) < 1$. Then the previous lemma implies that bias converges linearly, and variance is $O(\log(t)/t)$.

To argue the existence of $\mu > 0$, we will perform eigenvalue perturbation analysis using the Bauer-Fike theorem. Note that $A(\alpha, \mu, h) = A(\alpha, \mu_0, h) + \mu \Delta_A$ where

$$\Delta_A = \begin{pmatrix} 1 & -1 \\ 0 & 0 \end{pmatrix}.$$

Similarly, $D(\alpha, \mu, h) \approx D(\alpha, \mu_0, h) + \mu \Delta_D$ where

$$\Delta_D = \begin{pmatrix} 2(1 - \alpha h) & 0 & -2(1 - \alpha h) \\ 0 & 0 & 0 \\ 1 & 0 & -1 \end{pmatrix}.$$

This last approximate inequality is a first-order approximation, in the sense that we are working with arbitrarily small, positive values of $\mu$, and we have kept terms linear in $\mu$ but ignored higher order terms, like $\mu^2$.

We will apply the Bauer-Fike theorem to bound the eigenvalues of $D(\alpha, \mu, h)$. Consider the eigendecomposition $D(\alpha, \mu_0, h) = V \Lambda V^{-1}$. We can compute

$$V = \begin{pmatrix} 0 & 0 & \frac{1 - 2\alpha h + 3\alpha^2 h^2}{1 - \alpha h} \\ 0 & 1 & \frac{1}{1 - \alpha h} \\ 1 & 0 & 1 \end{pmatrix}$$

and

$$V^{-1} = \begin{pmatrix} \frac{1 - \alpha h}{1 - 2\alpha h + 3\alpha^2 h^2} & -\frac{1}{1 - 2\alpha h + 3\alpha^2 h^2} & 0 \\ 0 & 1 & 0 \\ 1 & 0 & 0 \end{pmatrix}.$$

Note that because we assume $\alpha < 2/(3h)$ we get $1 - \alpha h > 0$. Also, $1 - 2\alpha h + 3\alpha^2 h^2 > 0$ regardless of the choice of hyperparameters. This means that matrices $V$ and $V^{-1}$ are singular and of finite norm. The norm of $\Delta_D$ is also finite. The Bauer-Fike theorem states that, if $\nu$ is an eigenvalue of $D(\alpha, \mu_0, h)$, then there exists an eigenvalue $\lambda$ of $D(\alpha, \mu, h)$ such that

$$|\lambda - \nu| \le \|V\|_p \|V^{-1}\|_p \|\mu \Delta_D\|_p,$$

for any $p$-norm. Since by construction $|\nu| \le \rho(D(\alpha, \mu_0, h)) < 1$, the above means that there exists a sufficiently small, but strictly positive value of $\mu$, such that $\lambda < 1$. By repeating this argument for all pairs of eigenvalues, we get the stated result. The same argument can be repeated to prove the existence of a strictly positive $\mu$ such that $\rho(A(\alpha, \mu, h)) < 1$.

$\square$