[Reviews · NeurIPS 2019]

Reviewer 1



Update: I have carefully read the author's rebuttal. I maintain my rating (good submission, promising ideas, would be better and more compelling if the empirical results were stronger). Overall this is well-written and makes interest theoretical and conceptual contributions. It seems likely to spark future work along these lines. It also seems of some practical significant in that variance reduction techniques have generally not previously been shown to have much success in improving the training of deep learning methods. Whereas, Figure 3 indicates that IGT was able to significant improve the training of ResNet for CIFAR10, for example. Curiously, the authors only examined ITA-Adam for IMDb and MAML, and not also for CIFAR10 and ImageNet. Especially given they are suggesting that ITA is a good plug-replacement for gradients, it would be interesting to see whether ITA generally improved not only Heavy Ball (classical momentum) but also Adam, across all the tasks they examined. The absence of Adam-ITA for the conv net tasks (CIFAR and ImageNet) makes readers wonder if ITA does not work well for Adam for those cases, and if so why not. If Adam-ITA does not perform well for them, that would be interesting to examine, to better understand limitations of ITA, which are currently not examined very well in this work as is. That is, ITA seems to always be at least as good or better than the non-ITA base optimizers, but it is not clear if there is fundamental theoretical reasons that will always be the case -- especially since the impact of violating of the assumption of equal Hessians (even when addressed with anytime tail averaging) in practice is not clear or bounded in general.

Reviewer 2



Very well written paper. Clear statement of contribution. Few assumptions made about reader's existing knowledge of stochastic gradients and online learning. Good review of existing methods, which supports motivation of proposed method. The new method itself is introduced analytically with assumptions that don't hold in practice, but addressed experimentally. Thorough experimental details and proof in appendix. The overall research is sound and makes a novel contribution, which appears to advance the state of the art of some machine learning tasks.

Reviewer 3



This paper proposes a novel gradient estimator that performs a weighted average (similar to momentum) of past and new gradients to estimate the gradient at the current iterate. To motivate their estimator, the authors demonstrate that the SG method with momentum does not decrease the variance unless the momentum parameter is increased like 1 – 1/t. The IGT estimator is then derived by considering the quadratic case (where the Hessian matrix is fixed for all individual functions) with the goal of estimating the “true” online gradient (the simple average over all previously seen gradients). In order to compensate for the bias of past gradients, the new gradient is notably evaluated at an extrapolated point, not at the current point. This derived estimator yields an O(1/t) reduction in variance, yielding a theoretical result that may be interpreted as linear convergence to a neighborhood that shrinks as O(1/t) with constant steplength for quadratic problems. In order to generalize their method to non-quadratic settings, the IGT parameter is modified using anytime tail averaging to ensure only the last c fraction of gradients are weighted heavily. The method is demonstrated to be effective on image classification, sentiment analysis, reinforcement learning, and meta-learning applications. Strengths: This paper proposes a novel gradient estimator similar in form to momentum that yields variance reduction in the online and stochastic (non-finite sum) setting. This is significant, as up until this point, almost all variance reduction methods (such as SAG, SAGA, SVRG, etc.) rely on the finite support structure of the objective function. Albeit under a restricted setting (the quadratic case), the authors are able to derive a gradient estimator that reduces the variance as O(1/t) with the same cost as SGD at every iteration. This is impressive and seems to yield a useful practical algorithm when combined with anytime tail averaging. The paper is also well-written. Weaknesses: The primary concern I have regarding the algorithm is its analysis, which is currently restricted to the quadratic setting and is only able to attain an optimal convergence rate of O(1/t). Although the variance is reduced and the algorithm can converge with a constant steplength, the variance reduction is not sufficient to attain linear convergence as with other variance reduction algorithms (SAG, SAGA, SVRG, SARAH, etc.), and in fact yields the same convergence rate as SGD with a diminishing steplength on strongly convex functions. That being said, those other variance reduction methods explicitly exploit the finite-sum structure. This is the first approach (to my knowledge) where the variance is reduced in the online setting without the use of an increasing batch size. In addition, although having more general convergence guarantees would be nice, there are many other algorithms whose convergence can only be proven currently in the quadratic case that have empirically been shown to be useful and eventually generalized to more generic settings (for example, the conjugate gradient method and nonlinear acceleration). I had a few questions for the authors: 1. In order for the variance to be reduced, the IGT parameter (gamma) has to be increased by something like 1 – 1/t. (This also appears to be the case for anytime tail averaging.) This results in extrapolations that are increasingly far from the current iterate (in the original algorithm, by t, more generally (gamma / (1 – gamma)). Have the authors observed any potential numerical instabilities that could arise from this extrapolated evaluation? 2. Three different methods are discussed in this paper: (1) SGD with momentum, (2) IGT, and (3) deterministic heavyball. Although all of these methods are very similar in form, the authors (perhaps implicitly) imply that these methods are quite different in purpose (such as for variance reduction and/or curvature adaptation). Could the authors clarify on how these three methods ought to be distinguished? 3. Along these lines, the authors propose an algorithm combining heavy ball and IGT. We know that the heavy ball method does not necessarily converge in general for strongly convex functions. Why is it reasonable to apply IGT with heavy ball for neural networks? How is heavyball-IGT different or similar to momentum-SGD for neural networks? Is heavyball-IGT even necessary? (Heavyball-IGT seems to perform quite similarly to IGT in many of the experiments). 4. Lastly, how does IGT with a fixed gamma parameter compare against SGD with momentum with a fixed momentum parameter? 5. Have the authors considered comparing this method against other variance reduction methods, both for neural networks and perhaps a simpler problem, such as l2-regularized logistic regression? (To be clear, I do not think that this is not necessary, but it would be interesting to see.) An additional relevant work that is not cited in this paper is: - Defazio, Aaron, and Léon Bottou. "On the ineffectiveness of variance reduced optimization for deep learning." arXiv preprint arXiv:1812.04529 (2018). Typos: - Pg. 2, line 37: The variable N has not yet been introduced (although I assume it is the batch size) - Pg. 2, line 37: multiple, not multiple - Pg. 4, line 98: travelled, not travalled - Pg. 16, line 448 and 450: I would use a comma “,” instead of a “’” when writing the number of datapoints - Pg. 24, line 577: capitalize Hessian - Pg. 26, line 608: decreasing, not Decreasing - Pg. 26, equations below lines 610, 611, 615: should be max {1 + r_h, 2 log(2/(i (1 – r_h))) } Overall, this paper proposes a novel approach for performing variance reduction in the non-finite sum setting. The paper is well-written and gives clear evidence (both theoretically and experimentally) for their method. I believe that the ideas are significant as it could trigger research on other approaches for variance reduction in the online setting and takes a step towards understanding the purpose of momentum for SGD. Hence, the manuscript is deserving of publication. ======================================================== I am grateful for and satisfied with the authors' response and have kept my decision as an acceptance.

[Author Response · NeurIPS 2019]

We wish to thank all reviewers for the time they spent reading and commenting on our paper. We will amend the text with all the suggested clarifications and corrections.

We now reply to the most important concerns.

## Momentum (or Nesterov acceleration) and variance reduction

One of the questions raised by reviewer 3 is a crucial one. While momentum is proved to help with curvature, its formulation resembles that of variance reduction. The theorems in [39] show a greater sensitivity of acceleration to noise, although this might be a limitation of the analysis. After the NeurIPS deadline, the paper "Limitations of the Empirical Fisher Approximation" was published showing that using the covariance matrix of the gradients in lieu of the Fisher (which can be thought of as a curvature matrix) was problematic. The paper "Information matrices and generalization" shows that, although the covariance and the Hessian encode different pieces of information, they are remarkably aligned in the currently used models. In conclusion, although we view deterministic momentum as helping with curvature, IGT as helping with noise and stochastic momentum as a successful method without general theoretical guarantees, there seems to be deep links between variance and curvature and the discussion of which method addresses what problem is not over.

## Convergence rate of IGT

R3 mentions the O(1/t) rate. Although variance-reduction methods achieve a linear rate of convergence on the training error, the rate on the test error is still no better than O(1/t), the minimax rate. Since we are in the online setting, we are directly optimizing the test loss and thus cannot do better than O(1/t).

## Weaknesses of ITA

ITA interpolates between only using the last iterate, which is robust to a violation of the identical Hessian assumption but does not decrease the variance, and using all of them, which has the lowest variance but suffers when the assumption is violated. ITA gets the best of both worlds but this come at a price, albeit small. First, the variance is greater by a constant factor than when all the iterates are used. Second, there is an additional hyperparameter: the fraction of examples to keep. While we found that our method was competitive for a wide range of values and that setting it so that one full epoch was used at the end of training worked well in all cases, it has an impact nonetheless.

## Performance of Adam-ITA

To be true to our goal of limiting the number of hyperparameters of IGT, we used the same values for the parameters as the non-IGT version. While this worked well for most methods, Adam-ITA did not work well. We did not want to falsely give the impression that it cannot work well but the computational cost to fine-tune Adam-ITA parameters on these large datasets would have been prohibitive. We can however run these experiments if the reviewers think there is a lot of value in them.

## Clarification about the restricted setting

We would like to point to R3 that our theorems apply in a more restrictive setting than the functions being quadratic as we require all individual Hessians to be equal. It is unclear how to extend the theorems to the general quadratic case.

## Miscellaneous questions

We did not observe any numerical instabilities. Although the factor in the extrapolation step grows, subsequent iterates get closer so the extrapolation step is never too far away. HB-IGT indeed barely improves upon IGT



[Meta-Review · NeurIPS 2019]

This is very good paper, which has found unanimous support among the reviewers.